# Comparative Transcriptome Analysis of the Differential Effects of Florpyrauxifen-Benzyl Treatment on Phytohormone Transduction between Florpyrauxifen-Benzyl-Resistant and -Susceptible Barnyard Grasses (*Echinochloa crus-galli* (L.) P. Beauv)

Wenyong Jin, Jinqiu Sun [ID], Wei Tang, Yongjie Yang, Jianping Zhang, Yongliang Lu and Xiaoyue Yu *

State Key Laboratory of Rice Biology and Breeding, China National Rice Research Institute, Hangzhou 311401, China
* Correspondence: yuxiaoyue@caas.cn; Tel.: +86-0571-63370288

**Abstract:** *Echinochloa crus-galli* (L.) P. Beauv (common name: barnyard grass) is a major weed in rice-growing areas and has evolved resistance to multiple herbicides. Florpyrauxifen-benzyl (trade name Rinskor) is a novel synthetic auxin herbicide that was approved in China in 2017 and is widely used in rice production to control resistant weeds, including barnyard grass. We identified a florpyrauxifen-benzyl-resistant *E. crus-galli* biotype with a resistance index (RI) of 11.89 using screen house herbicide experiments. To understand the phytotoxicity mechanisms of florpyrauxifen-benzyl, we used transcriptomics technologies to compare the gene expression profiles of florpyrauxifen-benzyl treatment on phytohormone transduction between florpyrauxifen-benzyl-resistant and -susceptible barnyard grasses (*Echinochloa crus-galli* (L.) P. Beauv). A total of 1810 DEGs were identified in the S comparison setting (FTS vs. UTS), and 915 DEGs were identified in the R comparison setting (FTR vs. UTR); 464 genes overlapped between the two comparison groups. Approximately sixty-nine hormone-related DEGs were detected after treatment with florpyrauxifen-benzyl in both R and S biotypes. At 24 h after florpyrauxifen-benzyl treatment, compared with the R biotype, the S biotype showed a stronger auxin response and higher expression of related genes involved in ethylene and abscisic acid biosynthesis and signal transduction. In addition, a brassinolide receptor gene was upregulated after florpyrauxifen-benzyl treatment and had higher expression in the S biotype than in the R biotype. This study is the first transcriptome analysis of the differential effects of florpyrauxifen-benzyl treatment between florpyrauxifen-benzyl-resistant and -susceptible *E. crus-galli*. It reflects the difference in phytohormone biosynthesis and signal transduction between R and S barnyard grasses in response to florpyrauxifen-benzyl treatment and will be helpful for understanding the phytotoxicity mechanisms of florpyrauxifen-benzyl.

**Keywords:** florpyrauxifen-benzyl; barnyard grasses; phytohormone transduction; resistance



## 1. Introduction

*Echinochloa crus-galli* (L.) P. Beauv (common name: barnyard grass), an annual plant belonging to the Poaceae family and propagated via seeds, is a major weed in rice-growing areas [1]. Since the discovery of the first case of herbicide-resistant barnyard grass (*E. crus-galli*) in 1978 (Maryland, USA), 49 cases of herbicide-resistant barnyard grass (*E. crus-galli*) have been identified up to 2022, of which 37 cases were found in rice fields. These 37 cases of resistant barnyard grass found in rice fields included a total of 8 Herbicide Resistance Action Committee (HRAC) groups, including HRAC group A (cyhalofop-butyl, fenoxaprop-ethyl, and quizalofop-ethyl), HRAC group B (foramsulfuron, imazapyr, nicosulfuron, and penoxsulam), HRAC group C (propanil), HRAC group F4 (clomazone), HRAC group K3

(butachlor), HRAC group L (quinclorac, with a mechanism of action in monocots), HRAC group N (thiobencarb), and HRAC group O (quinclorac) [2].

Florpyrauxifen-benzyl (trade name Rinskor) is a broad-spectrum, synthetic auxin herbicide (HRAC group O) that was approved in China in 2017. It is an innovative aryl-pyridine methyl ester herbicide developed by Dow AgroSciences and is used as a post-emergence herbicide in rice production. Synthetic auxin herbicides (SAHs) primarily mimic excessive amounts of the natural auxin indole-3-acetic acid (IAA), which disrupts the homeostasis of endogenous plant hormones and severely impairs plant growth and development, eventually resulting in plant death [3]. However, the exact mechanism of action of SAHs remains unclear.

According to the classical model of the auxin signaling pathway, the excess IAA binds to the auxin receptor protein complex $SCF^{TIR/AFB}$-Aux/IAA (SCF refers to SKP1-Cullin-Fbox, TIR1/AFB refers to Transport Inhibitor Resistant 1/Auxin Signaling F-Box, Aux/IAA refers to Auxin/Indole Acetic Acid) and promotes ubiquitination of the Aux/IAA proteins; then, the ubiquitination-tagged Aux/IAA proteins will be degraded by the 26S proteasome to release auxin response factor (ARF), which affects the expression of downstream genes to produce the auxin responses [4–7]. Multiple members of the auxin receptor family, such as TIR1, AFB1, AFB2, AFB3, AFB4, and AFB5, have been identified in arabidopsis, and there is functional redundancy among the different auxin receptor proteins [8–10]. Different synthetic auxin herbicides have been found to have different affinities for different auxin receptor proteins, such as previous synthetic auxin herbicides (e.g., quinclorac, 2,4-dichlorophenoxyacetic acid), which have an affinity for TIR1; however, florpyrauxifen-benzyl prefers the AFB5 IAA coreceptor to the TIR1 coreceptor as its site of action [11–14]. This discrepancy allows florpyrauxifen-benzyl to be used to control barnyard grass that has developed resistance to quinclorac, a synthetic growth hormone herbicide widely used in the past [15].

Recently, florpyrauxifen-benzyl has rapidly become an important tool for controlling barnyard grass in rice production because of its novel action site, good herbicidal effect, and safety in rice. A Korean study evaluated the baseline sensitivity of *Echinochloa* species to florpyrauxifen-benzyl, showing that $GR_{50}$ values for *E. oryzicola* ranged from 4.54 g to 29.66 g a.i. $ha^{-1}$, whereas those for *E. crus-galli* ranged from 6.15 g to 16.06 g a.i. $ha^{-1}$ [16]. Other investigators have also demonstrated that florpyrauxifen-benzyl may be used with imazethapyr and that the addition of malathion does not increase the risk of damage to rice [17,18].

The frequent use of florpyrauxifen-benzyl has led to the resistance of barnyard grass to it. Therefore, there is a need to understand the phytotoxicity and resistance mechanisms of florpyrauxifen-benzyl. Part of the phytotoxicity mechanism of florpyrauxifen-benzyl and the non-target-site resistance mechanism of barnyard grass to florpyrauxifen-benzyl has have recently been reported on. A study based on phytohormone content and antioxidant enzyme activities suggested that ethylene biosynthesis stimulation, abscisic acid buildup, and reactive oxygen species (ROS) play essential roles in the mechanism of florpyrauxifen-benzyl phytotoxicity in barnyard grass [19]. Enzymatic hydrolysis converts the benzyl ester present in the applied form of florpyrauxifen-benzyl to its active acid form, florpyrauxifen acid [20,21]. Several previous studies have found differences in florpyrauxifen-benzyl metabolism between sensitive and resistant barnyard grass biotypes, as evidenced by the content of the active metabolite florpyrauxifen acid being greater in sensitive barnyard grass than in resistant barnyard grass, suggesting that reductions in florpyrauxifen-benzyl absorption and florpyrauxifen acid production may contribute to the inability to control barnyard grass with florpyrauxifen-benzyl [22]. Furthermore, the evolution of florpyrauxifen-benzyl resistance in multiple-resistant barnyard grass can be linked to non-target-site resistance mechanisms that reduce herbicide absorption and translocation and cause reduced conversion or rapid degradation of florpyrauxifen acid [23].

Plant responses to synthetic auxin herbicides involve a vast and intricate network of regulatory processes including phytohormone homeostasis, signal transduction, photo-

synthesis, and material metabolism. With the development of genomics, transcriptomic profiling, particularly RNA-Seq analysis, is becoming a more frequent tool for studying plant–herbicide interactions and is widely used to study herbicide resistance and the effects of herbicides on plant growth and development [24–26]. Although some transcriptome analyses of other SAHs have been reported, transcriptome studies on florpyrauxifen-benzyl are still lacking [26,27]. The effects of florpyrauxifen-benzyl on resistant and susceptible barnyard grasses and the differences between these effects have also been poorly studied. We hypothesized that there is a difference between the effects of florpyrauxifen-benzyl on resistant and susceptible barnyard grasses and analyzed this difference using transcriptomic approaches. This study aimed to clarify the differential effects of florpyrauxifen-benzyl treatment between florpyrauxifen-benzyl-resistant and -susceptible barnyard grasses by comparing the gene expression profiles of florpyrauxifen-benzyl herbicide-resistant and -susceptible barnyard grasses (*E. crus-galli*) after florpyrauxifen-benzyl treatment.

## 2. Materials and Methods

### 2.1. Plant Materials and Growth Conditions

#### 2.1.1. Plant Material Description

The resistant *E. crus-galli* accession was found near a grower's field in Yuanjiang County, Hunan Province, China, in 2017, whereas the susceptible *E. crus-galli* accession was found in a grower's field in Wangjiang County, Anhui Province, China, in 2018. The G2 seeds for each biotype were harvested separately from resistant (R) and susceptible (S) plants grown in screen houses under natural conditions. Seeds were harvested in May 2019, dried at 35 °C for a week, and kept in the dark at 4 °C until use. Bromothymol blue was used to measure the seed viability, which was approximately 100%.

#### 2.1.2. Dose–Response Experiments of Florpyrauxifen-Benzyl

During the pre-screening experiment, we found that the R biotype was resistant to florpyrauxifen-benzyl. To determine the $GR_{50}$ (the herbicide dose that inhibits plant growth by 50%) of biotypes R and S for florpyrauxifen-benzyl, we performed dose–response experiments with florpyrauxifen-benzyl. The experiments were conducted from May to July 2020 in a screen house (35 °C/20 °C day/night temperature with natural sunlight) at the China National Rice Research Institute (30.08′ N, 119.94′ E). Approximately 20 seeds were sown on the soil surface in plastic pots (7.5 cm diameter) and covered with a thin layer of sterilized soil. At about a week after sowing, the seedlings were thinned into three plants per container. Plants were treated with various doses of florpyrauxifen-benzyl at the 3–4 leaf stage, and untreated barnyard grasses were planted under the same conditions. The aerial parts of each potted plant were harvested 14 days after treatment to determine the fresh biomass. To standardize comparisons between populations, the fresh weight was expressed as a percentage of the untreated control. Three pots per treatment were used as replicates, and the experiment was repeated three times. During the entire experiment, the soil was kept moist but not flooded by the daily addition of water (for approximately 30 days).

Florpyrauxifen-benzyl was applied using a laboratory spray tower (model 3WP-2000) manufactured by the Nanjing Institute of Agricultural Mechanization, Ministry of Agriculture, China, equipped with a flat fan nozzle (TP6501E), delivering 200 L ha$^{-1}$ at 230 kPa. The recommended field dose of florpyrauxifen-benzyl is 18 to 36 g a.i. ha$^{-1}$, and according to the different performances of biotypes R and S in the initial evaluation, we set the treatment doses for biotypes R and S separately in the florpyrauxifen-benzyl resistance test (Table 1).

**Table 1.** The application dose of florpyrauxifen-benzyl used in the dose–response experiment.

| Biotype | Dose (g a.i. ha$^{-1}$) | | | | | | | |
|---|---|---|---|---|---|---|---|---|
| Resistant (R) | 1.875 | 3.75 | 7.5 | 15 | 30 | 60 | 120 | 240 |
| Susceptible (S) | 0.39 | 0.78 | 1.56 | 3.13 | 6.25 | 12.5 | 25 | 50 |

### 2.1.3. Seedling Preparation for Transcriptome Analysis

The G2 seeds used for transcriptome analysis of each biotype (resistant and susceptible) were planted in 7.5 cm diameter plastic pots (approximately 20 seeds per pot). The seedlings were grown in the same screen house at the China National Rice Research Institute under growth conditions similar to those used in the dose–response experiments of florpyrauxifen-benzyl. Treatments and sampling were conducted at the 3–4 leaf stage of the weeds. Old leaves of resistant and susceptible biotypes that were untreated and florpyrauxifen-benzyl-treated (30 g a.i. ha$^{-1}$, 24 h after treatment) were collected for transcriptome analysis.

### 2.2. Statistical Treatment of Dose–Response Curves

The florpyrauxifen-benzyl dose–response experiments were conducted in a randomized complete block experiment with three replicates for each biotype and herbicide treatment, and the experiment was repeated three times. In the florpyrauxifen-benzyl dose–response experiments, a three-parameter logistic equation was fitted to the weed biomass of R and S biotypes under florpyrauxifen-benzyl treatment in combination with florpyrauxifen-benzyl dose using the "drc" add-on package in R 3.5.3 (R Core Team, 2019):

$$Y = d/[1 + (x/GR_{50})^b], \tag{1}$$

where $Y$ refers to the fresh weight expressed as a percentage of the control; $d$ refers to the upper limit of fresh weight at dose zero; $b$ refers to the slope at $GR_{50}$; and $GR_{50}$ refers to the herbicide dose required for 50% growth reduction. The resistance index (RI) was calculated as the mean of the $GR_{50}$ of the R biotype divided by that of the S biotype to determine the level of resistance of the *E. crus-galli* biotypes.

### 2.3. RNA Isolation and Transcriptome Sequencing

Extraction of total RNA from the leaves of *E. crus-galli* of control (untreated) and florpyrauxifen-benzyl-treated plants (30 g a.i. ha$^{-1}$, 24 h after treatment) and RNA-Seq were performed by Novogene (Beijing, China). Three biological replicates were used for each sample: UTR_1, UTR_2, UTR_3, UTS_1, UTS_2, UTS_3, FTR_1, FTR_2, FTR_3, FTS_1, FTS_2, and FTS_3. In sample naming, UT refers to untreated and FT refers to florpyrauxifen-benzyl treated, R refers to resistant *E. crus-galli*, and S refers to susceptible *E. crus-galli*. Novogene used an RNA prep Pure Plant Kit to extract RNA from *E. crus-galli* leaves (QIAGEN, Germany), and RNA samples were subjected to rigorous quality control using the Agilent Bioanalyzer 2100 system (Agilent Technologies, Carlsbad, CA, USA) to accurately determine the integrity of total RNA. All samples were examined for RNA degradation and contamination on 1% agarose gel. The purity of total RNA was measured using a NanoPhotometer® spectrophotometer (IMPLEN, Munich, Germany). From each sample, 1.5 μg RNA was used as input material for sample preparation. Sequencing libraries were generated using a NEBNext® Ultra™ RNA Library Prep Kit for Illumina® (NEB, Ipswich, MA, USA), according to the manufacturer's instructions.

### 2.4. Transcriptome Assembly and Functional Classification

Clean reads were obtained as follows: (1) the adapter-containing reads were removed, (2) the reads containing N base were deleted, and (3) the low-quality reads with more than 50% low-quality bases ($Q_{phred} \leq 20$) were eliminated. Meanwhile, the clean data's Q20, Q30, GC-content, and sequence duplication levels were determined. All downstream analyses were based on clean, high-quality readings. To build the full-length transcript sequences,

all clean reads from the 12 libraries were collected using Trinity software (v2.4.0) [28]. The transcriptome obtained by Trinity splicing was used as the reference sequence (Ref), and the clean reads of each sample were mapped to the Ref by RSEM software (v1.2.15), which uses the bowtie2 parameter mismatch 0 (the default bowtie2 parameter) [29]. The following databases were used to annotate gene function: NR (NCBI non-redundant protein sequences); NT (NCBI non-redundant nucleotide sequences); Pfam (protein family); KOG/COG (clusters of orthologous groups of proteins); Swiss-Prot (a manually annotated and reviewed protein sequence database); KO (KEGG ortholog database); GO (gene ontology database) (gene ontology). The transcription factors were discovered using iTAK 1.2 [30].

### 2.5. Differential Gene Expression and Enrichment Analysis

Gene expression levels were inferred using RSEM software (v1.2.15) with the fragments per kilobase of exon per million fragments mapped (FPKM) method [29,31]. The DESeq2 R package (Version 3.0.3) was used to analyze the differential expression of S and R in florpyrauxifen-benzyl-treated (FT) versus untreated (UT) samples [32]. DESeq2 provides statistical routines to determine differential expression in digital gene expression data using a model based on the negative binomial distribution. The resulting $p$-values were adjusted using the approach of Benjamini and Hochberg to control for the false discovery rate. Genes with an adjusted $p$-value < 0.05 and $|\log2 (\text{fold change})| > 1$ found by DESeq2 were classified as differentially expressed. All differentially expressed genes (DEGs) were enriched in the Gene Ontology (GO) and Kyoto Encyclopedia of Genes and Genomes (KEGG) databases. Gene Ontology (GO) enrichment analysis of differentially expressed genes was performed using the R package clusterProfiler (Version 3.0.3), in which gene length bias was corrected. GO terms with a corrected $p$-value of less than 0.05 were considered significantly enriched by differentially expressed genes. KEGG is a database resource for understanding the high-level functions and utility of a biological system, such as the cell, organism, and ecosystem, based on molecular-level information, particularly large molecular datasets generated by genome sequencing and other high-throughput experimental technologies (http://www.genome.jp/kegg/ (accessed on 1 December 2021)). We used the R package clusterProfiler (Version 3.0.3) to test the statistical enrichment of genes with differential expression in KEGG pathways [33].

### 2.6. RT–qPCR Validation

To confirm the reliability of the transcriptome sequencing results, the DEGs' transcriptome-level transcripts were examined using RT–qPCR. Total RNA was extracted from the UTR, UTS, FTR, and FTS leaves using the RNAprep Pure Plant Extraction Kit under conditions identical to those used for RNA-Seq (Tiangen, China). Then, 1 μg of the RNA sample was reverse-transcribed using a cDNA synthesis kit (TAKARA, China). RT–qPCR was performed on a QuantStudio™ 1 real-time PCR system using an iTaq Universal SYBR Green Supermix (Bio-Rad, Hercules, CA, USA) under the following thermal cycle conditions: denaturation at 95 °C for 30 s, followed by 40 cycles of 95 °C for 5 s and 60 °C for 1 min. The coding sequences (CDSs) of the selected genes were used to design specific primers for RT–qPCR (Supplementary Table S5). Actin was used as the housekeeping gene [34]. Each treatment was replicated three times, and the average data from three technical replicates were obtained using the $2^{-\Delta\Delta Ct}$ method to calculate the relative expression levels [35].

## 3. Results

### 3.1. Florpyrauxifen-Benzyl Efficacy against Resistant and Susceptible E. crus-galli

The $GR_{50}$ value of S was 2.97 g a.i. ha$^{-1}$, and the $GR_{50}$ value of R was 35.30 g a.i. ha$^{-1}$. Dose–response curves showed that the R biotype was resistant to florpyrauxifen-benzyl (Figure 1), with a resistance index (RI) of 11.89.

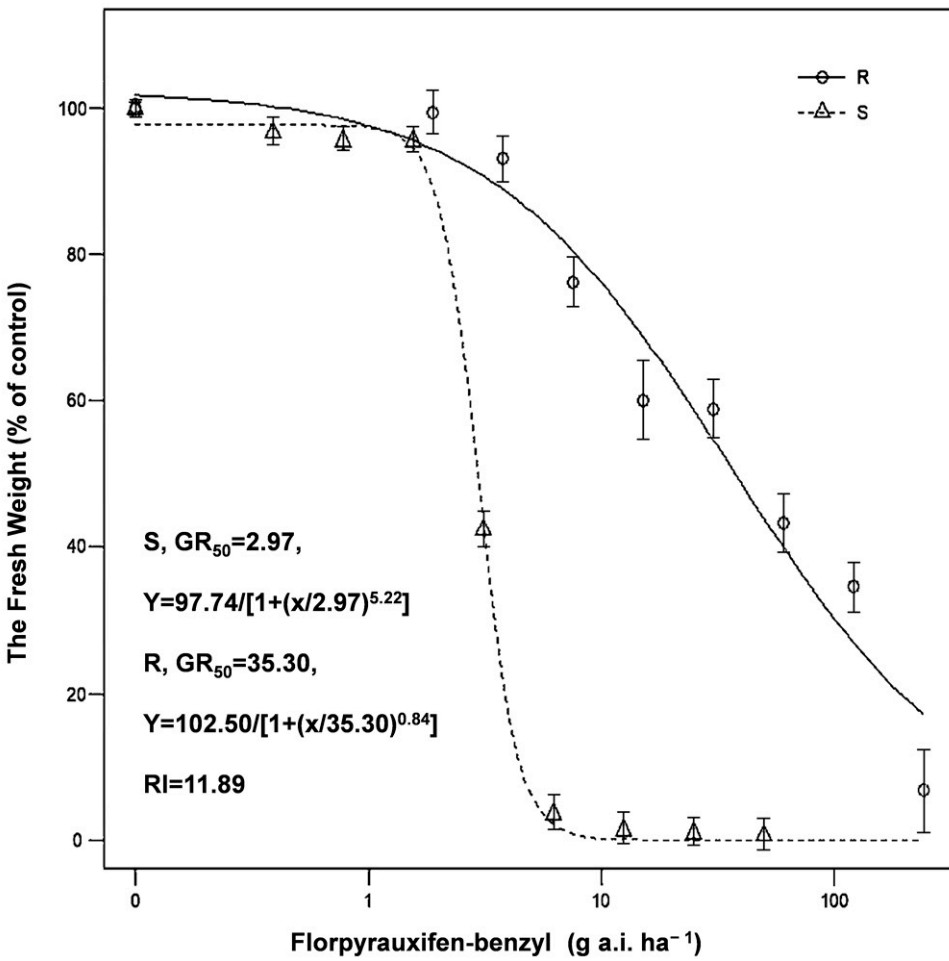

**Figure 1.** Dose-response curves to florpyrauxifen-benzyl of S and R *E. crus-galli* biotypes.

*3.2. De Novo Assembly of E. crus-galli Reference Transcriptome*

The RNA-Seq analysis included 12 samples. The total number of clean reads per sample is listed in Supplementary Table S1, and the number of annotated unigenes per database is listed in Supplementary Table S2. Essentially, 62,145 genes (68.61%) were annotated in at least one database, of which 47,434 (52.37%) were annotated in the NR database, indicating that the de novo transcriptome possessed relatively complete information on gene function.

*3.3. Identification of Differentially Expressed Genes*

To identify the plant response after florpyrauxifen-benzyl treatment in florpyrauxifen-benzyl herbicide-resistant and -susceptible *E. crus-galli*, two comparison settings (FTS vs. UTS and FTR vs. UTR) were analyzed. Volcano plots directly display the distribution of DEGs in R and S (Figure 2A,B). The high correlation values showed that the reproducibility of all the biological samples was good (Figure 2C). A total of 1810 DEGs were identified in the S comparison setting (FTS vs. UTS), and 915 DEGs were identified in the R comparison setting (FTR vs. UTR); 464 genes overlapped between the two comparison groups, as depicted in the Venn diagram (Figure 2D).

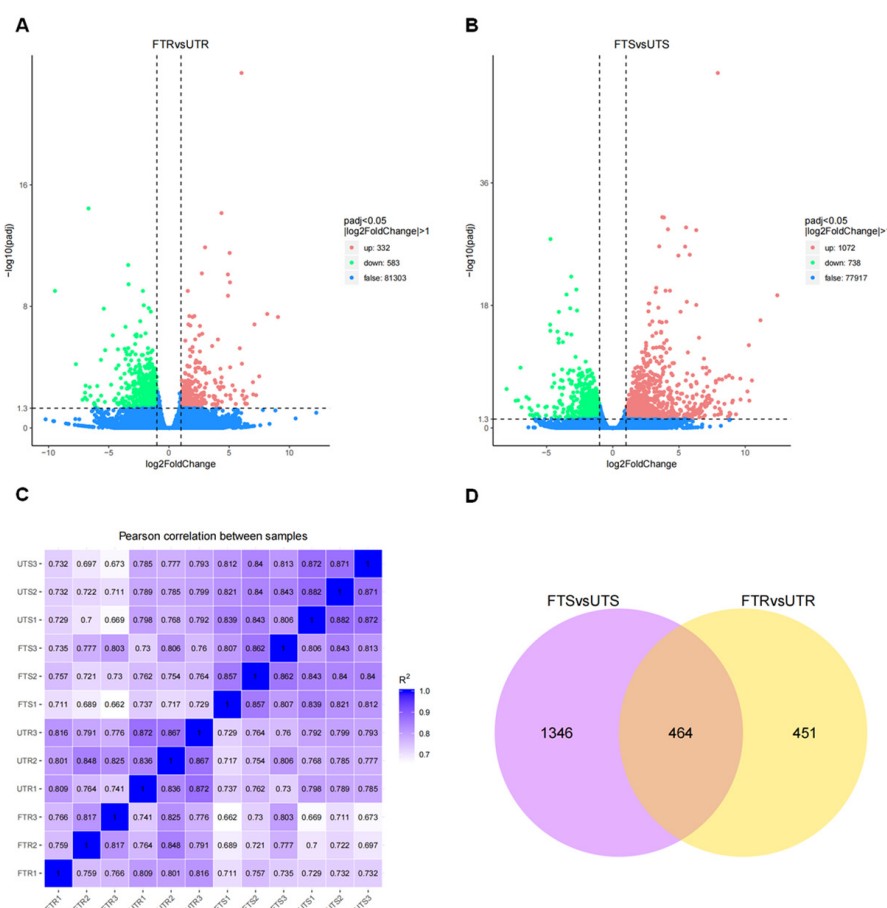

**Figure 2.** Analysis of differentially expressed genes (DEGs). (**A**) Volcano plot of DEGs in the S comparison setting (FTS vs. UTS). (**B**) Volcano plot of DEGs in the R comparison setting (FTR vs. UTR). (**C**) Correlation between the transcriptomes of the three biological replicates of FTS, UTS, FTR, and UTR. (**D**) Venn diagram of DEGs between the S comparison setting (FTS vs. UTS) and the R comparison setting (FTR vs. UTR).

*3.4. Enriched GO Terms and KEGG Pathways of DEGs*

To understand the different plant responses after florpyrauxifen-benzyl treatment in the florpyrauxifen-benzyl herbicide-resistant and -susceptible *E. crus-galli*, all DEGs were annotated against the GO and KEGG databases.

In the R comparison setting (FTR vs. UTR), 103 GO terms were enriched in upregulated DEGs, and 113 GO terms were enriched in downregulated DEGs (Supplementary Table S3). In the S comparison setting (FTS vs. UTS), 121 GO terms were enriched in upregulated DEGs, and 118 GO terms were enriched in downregulated DEGs (Supplementary Table S4). The top enriched GO terms of the upregulated and the downregulated DEGs in the R and S comparison settings were comparable, such as "oxidoreductase activity", "DNA-binding transcription factor activity", "photosynthesis", and "structural constituent of ribosome".

Similarly, all DEGs in the R and S comparison settings were enriched in the KEGG database. Figure 3A,B indicate the top 20 enriched pathways of upregulated DEGs in the R and S comparison settings; six of these pathways were the same in the R and S comparison settings. The bulk of upregulated DEGs in R and S biotypes were associated with "valine, leucine, and isoleucine degradation", "tryptophan metabolism", and "plant hormone signal transduction".

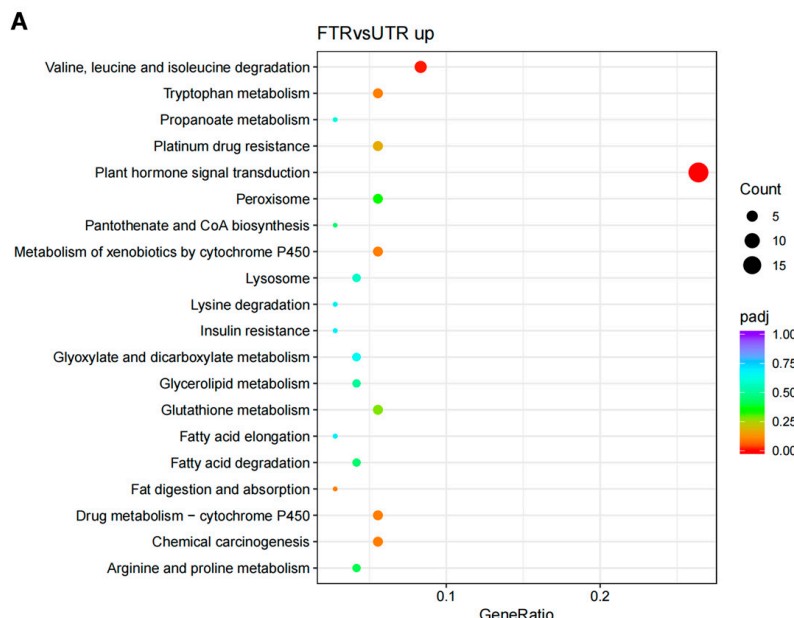

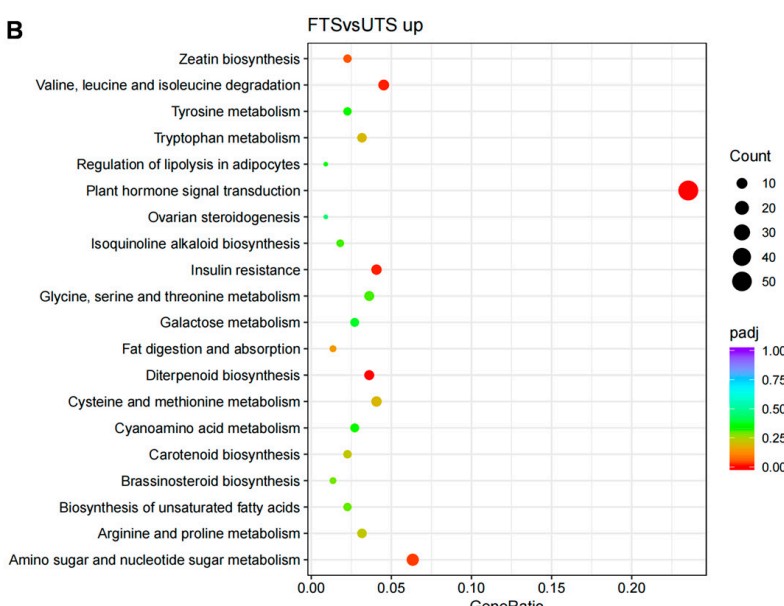

**Figure 3.** KEGG enrichment analysis of the upregulated DEGs. (**A**) Bubble chart of the top 20 enriched pathways of upregulated DEGs in the R comparison setting. (**B**) Bubble chart of the top 20 enriched pathways of upregulated DEGs in the S comparison setting. The size of the bubble indicates the number of DEGs, and the shaded color represents the Padj value.

Figure 4A,B show the top 20 enriched KEGG pathways of downregulated DEGs in R and S biotypes, of which 14 pathways were the same in both. The bulk of the downregulated DEGs were associated with "valine, leucine, and isoleucine biosynthesis", "ribosome", "porphyrin and chlorophyll metabolism", "photosynthetic-antenna proteins", and "photosynthesis".

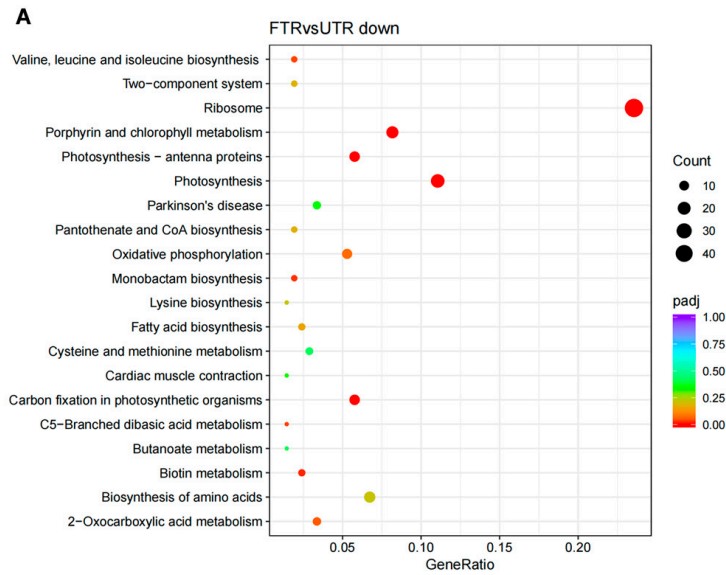

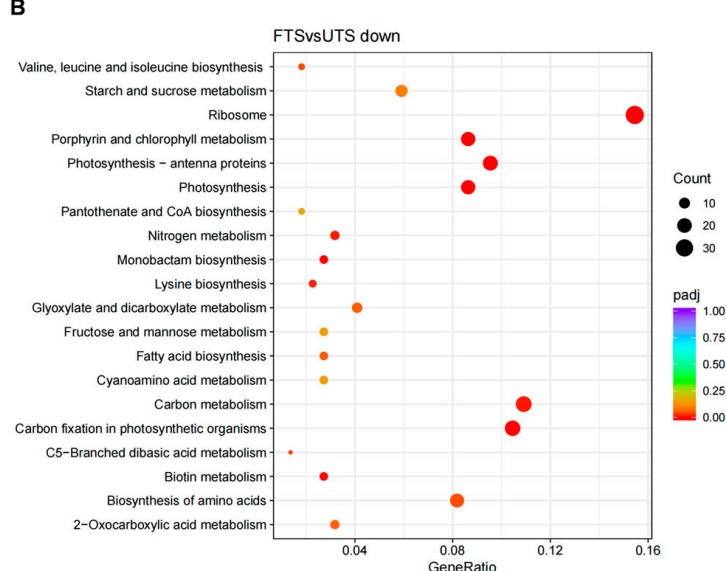

**Figure 4.** KEGG enrichment analysis of the downregulated DEGs. (**A**) Bubble chart of the top 20 enriched pathways of downregulated DEGs in the R comparison setting. (**B**) Bubble chart of the top 20 enriched pathways of downregulated DEGs in the S comparison setting. The size of the bubble indicates the number of DEGs, and the shaded color represents the Padj value.

### 3.5. DEGs Related to Phytohormone Biosynthesis and Signal Transduction

Florpyrauxifen-benzyl is a synthetic auxin herbicide that mimics the biological activity of endogenous auxins. Auxin has wide-ranging effects on plant growth and development and also interacts with other hormones. Therefore, we investigated the changes in gene expression related to plant hormone biosynthesis and signal transduction.

Approximately sixty-nine hormone-related DEGs were detected after treatment with florpyrauxifen-benzyl in both R and S biotypes (Supplementary Table S6). Most of these genes belonged to the auxin response pathway, followed by ABA (abscisic acid), ETH (ethylene), and CTK (cytokinin); relatively fewer genes were related to GA (gibberellic acid) and BR (brassinolide). Most of these genes showed increased expression, especially the auxin, ABA, and ETH pathway-associated genes, which were commonly induced by florpyrauxifen-benzyl treatment. Although these genes were upregulated in both R and S

biotypes, they differed in the degree of upregulation, and the increase was more significant in the S comparison setting.

### 3.6. Validation of DEGs by Real-Time Quantitative PCR

To confirm the accuracy and repeatability of the transcriptome data, thirty-two DEGs were selected to verify the RNA-Seq data using real-time quantitative PCR (RT–qPCR) (Table 2). Most of these genes showed increased expression, especially the auxin, ABA, and ETH pathway-associated genes, which were similar to the transcriptome results.

**Table 2.** The comparison between the transcriptome result and qPCR data for selected genes. ALDH: aldehyde dehydrogenase; AAO1_2: indole-3-acetaldehyde oxidase 1_2; IAA: auxin-responsive protein IAA; GH3: Gretchen Hagen 3; SAUR: small auxin upregulated RNA; NCED: 9-cis-epoxy carotenoid dioxygenase; ABF: ABA-responsive element (ABRE)-binding factors; PP2C: protein phosphatase 2C; SNRK2: sucrose non-fermenting 1-related protein kinases 2; ACS: 1-aminocyclopropane-1-carboxylate synthase; ACO: 1-aminocyclopropane-1-carboxylate oxidase; ETR, ERS: ethylene receptor and ethylene response sensor; EBF1_2: EIN3-binding F-box protein 1_2; ERF1: ethylene-responsive transcription factor; BRI1: brassinosteroid insensitive 1; AHK2_3_4: histidine kinase cytokinin receptors; AHP: arabidopsis histidine phosphotransfer proteins; ARR-A: type-A arabidopsis response regulators; CKX: cytokinin oxidases/dehydrogenases.

| Pathways | KO Name | Gene ID | Transcriptome log2FC * (FTR vs. UTR) | Transcriptome log2FC(FTS vs. UTS) | RT–qPCR log2FC(FTR vs. UTR) | RT–qPCR log2FC(FTS vs. UTS) |
|---|---|---|---|---|---|---|
| Auxin biosynthesis | ALDH | Cluster-7380.24619 | 2.20 | 3.15 | 1.33 | 3.05 |
| | AAO1_2 | Cluster-7380.23124 | 1.62 | 2.46 | 1.02 | 1.93 |
| Auxin signaling | IAA | Cluster-7380.10304 | 2.99 | 7.78 | 5.20 | 8.99 |
| | IAA | Cluster-7380.17655 | 1.53 | 2.99 | 0.88 | 2.70 |
| | IAA | Cluster-7380.23356 | 1.54 | 3.45 | 3.24 | 3.53 |
| | IAA | Cluster-7380.32469 | 1.78 | 2.79 | 0.90 | 2.13 |
| | IAA | Cluster-7380.33766 | 1.68 | 1.52 | −0.11 | 1.94 |
| | IAA | Cluster-7380.38071 | 1.34 | 1.71 | 0.79 | 1.80 |
| | IAA | Cluster-7380.54703 | 1.42 | 4.53 | 1.07 | 3.46 |
| | GH3 | Cluster-7380.9345 | 8.16 | 12.43 | 6.77 | 10.19 |
| | GH3 | Cluster-7380.9346 | 6.06 | 9.03 | 4.04 | 8.13 |
| | SAUR | Cluster-7380.2810 | 0.30 | 3.47 | −2.18 | 3.37 |
| | SAUR | Cluster-7380.15012 | 4.90 | 3.10 | 3.74 | 2.76 |
| ABA biosynthesis | NCED | Cluster-7380.40040 | 1.86 | 2.68 | 0.33 | 3.40 |
| ABA signaling | ABF | Cluster-7380.34634 | 1.59 | 1.95 | 0.50 | 1.60 |
| | PP2C | Cluster-7380.45755 | 1.54 | 1.77 | 1.03 | 2.09 |
| | PP2C | Cluster-7380.827 | 3.11 | 5.79 | 1.52 | 3.22 |
| | SNRK2 | Cluster-7380.20927 | 1.60 | 2.83 | 0.10 | 3.44 |
| ETH biosynthesis | ACS | Cluster-7380.61938 | 3.82 | 7.29 | 3.20 | 6.78 |
| | ACO | Cluster-7380.43266 | 1.42 | 5.05 | 1.66 | 4.15 |
| | ACO | Cluster-7380.43267 | 2.87 | 5.96 | 1.38 | 4.08 |
| ETH signaling | ETR, ERS | Cluster-7380.47127 | 0.49 | 1.09 | −0.34 | 1.48 |
| | EBF1_2 | Cluster-7380.29178 | 0.76 | 2.43 | −0.33 | 3.34 |
| | ERF1 | Cluster-7380.8146 | 1.45 | 8.78 | −0.52 | 6.40 |
| BR signaling | BRI1 | Cluster-7380.42410 | 0.57 | 1.28 | −0.48 | 0.95 |
| CTK signaling | AHK2_3_4 | Cluster-7380.32329 | −1.31 | −1.34 | −2.64 | −2.29 |
| | AHP | Cluster-7380.62034 | −2.01 | −0.67 | −3.42 | −0.18 |
| | ARR-A | Cluster-7380.31589 | −2.10 | −1.92 | −3.54 | −2.19 |
| | ARR-A | Cluster-7380.34172 | −1.82 | −1.38 | −2.06 | −2.42 |
| | ARR-A | Cluster-7380.40603 | −1.16 | −1.16 | −2.62 | −1.84 |
| | ARR-A | Cluster-7380.40690 | −2.61 | −3.16 | −3.97 | −2.26 |
| CTK biosynthesis | CKX | Cluster-7380.18387 | −3.68 | −2.08 | −4.54 | −1.78 |

* FC refers to fold change; the negative values of log2FC indicate downregulation.

To clarify the correlation between the transcriptome and RT–qPCR results, scatter plots were obtained using log2 fold variation measurements. The scatter plot showed that the

expression profiles of these DEGs were consistent with the RNA-Seq results, with relative $R^2 = 0.8722$ (Figure 5A) and $R^2 = 0.9331$ (Figure 5B) for R and S biotypes, respectively.

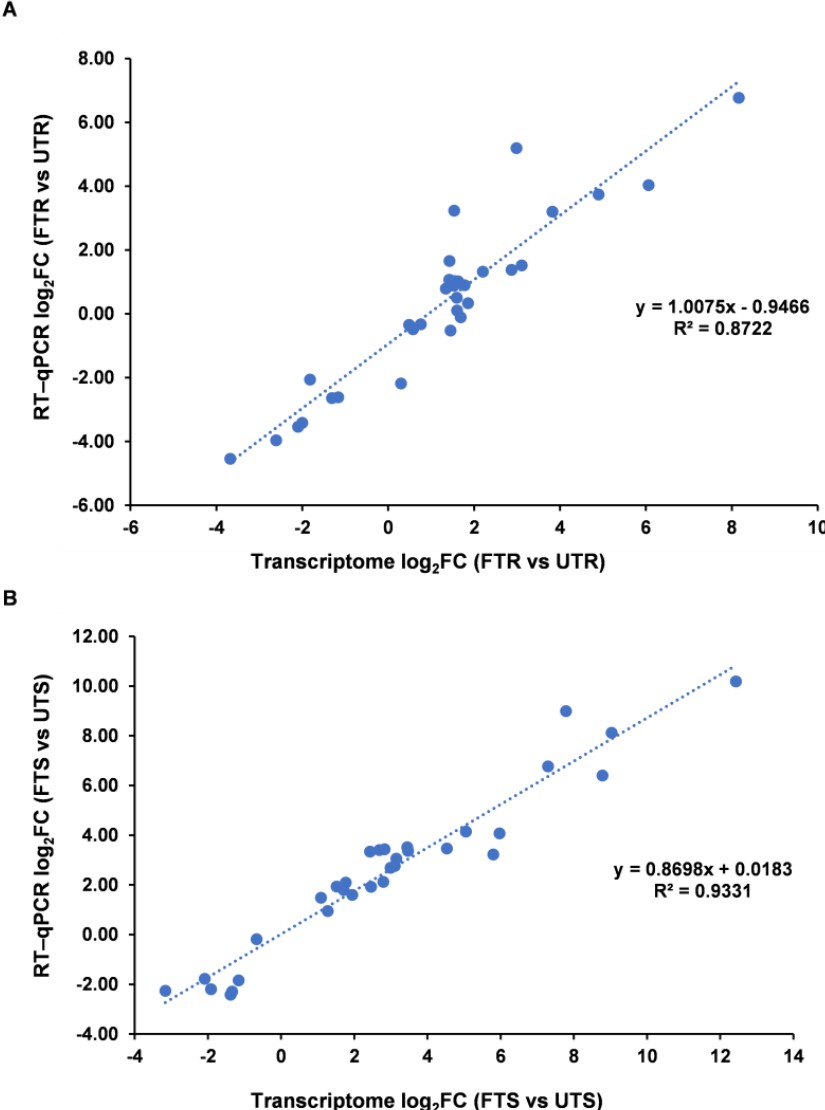

**Figure 5.** Scatter plots show the respective correlations. (**A**) The correlation between the transcriptome result and qPCR data of selected genes in the resistant biotype. (**B**) The correlation between the transcriptome result and qPCR data of selected genes in the susceptible biotype.

## 4. Discussion

We found that the pathways enriched by KEGG enrichment analysis of the differentially expressed genes in the S and R comparison settings had some degree of similarity, although they were not identical. Among the upregulated pathways, the plant hormone signal transduction pathway was the most significant and had the highest number of genes, whereas among the downregulated pathways, the ribosome pathway was the most significant and had the highest number of genes. In contrast to the R comparison setting, the expression of zeatin, carotenoid, and brassinosteroid biosynthesis pathways was upregulated in the S comparison setting. Zeatin, carotenoids, and brassinosteroids are the synthetic precursors of some phytohormones, and these pathways are related to plant hormone signal transduction. Therefore, it is necessary to analyze the differences in phytohormone signaling pathways in susceptible barnyard grass.

We present the biosynthesis and signal transduction pathways of some major phytohormones and the expression levels of related genes in the S and R comparison settings

in Figure 6. We observed that florpyrauxifen-benzyl-treated barnyard grass appeared to produce a stronger auxin response in the S than in the R biotype. The transcriptome results at 24 h after florpyrauxifen-benzyl treatment showed that the fold change (FC) in gene expression in the auxin-responsive GH3 gene family, auxin-responsive protein IAA, and SAUR family protein genes under the IAA signaling pathway was generally higher in S than in the R biotype (Supplementary Table S6). The intense auxin response is the result of high IAA concentrations upstream of the auxin signaling pathway, whereas the different IAA concentrations in SAH-treated plants are related to the absorption of SAHs by plants or the translocation and degradation of SAHs after absorption, on the one hand, and the IAA biosynthesis pathways, on the other hand [22,23]. Indole-3-acetaldehyde oxidase (AAO) is an important enzyme in the IAA biosynthesis pathways that oxidizes indole-3-acetaldehyde to indole-3-acetic acid (IAA), and oxidized indole-3-acetaldehyde is an intermediate in the tryptophan-dependent indole-3-pyruvic acid (IPA) auxin biosynthetic pathway [36,37]. Our transcriptome results showed that, after 24 h of florpyrauxifen-benzyl treatment, a gene encoding indole-3-acetaldehyde oxidase (Cluster-7380.23124) was significantly upregulated in both FTR and FTS compared with the untreated control (UTR and UTS). The gene was significantly more upregulated in FTS vs. UTS compared with FTR vs. UTR barnyard grass. Accordingly, we speculate that florpyrauxifen-benzyl treatment induced the activation of the indole-3-pyruvate (IPA) auxin biosynthesis pathway in barnyard grass and that S was more sensitive to this activation response.

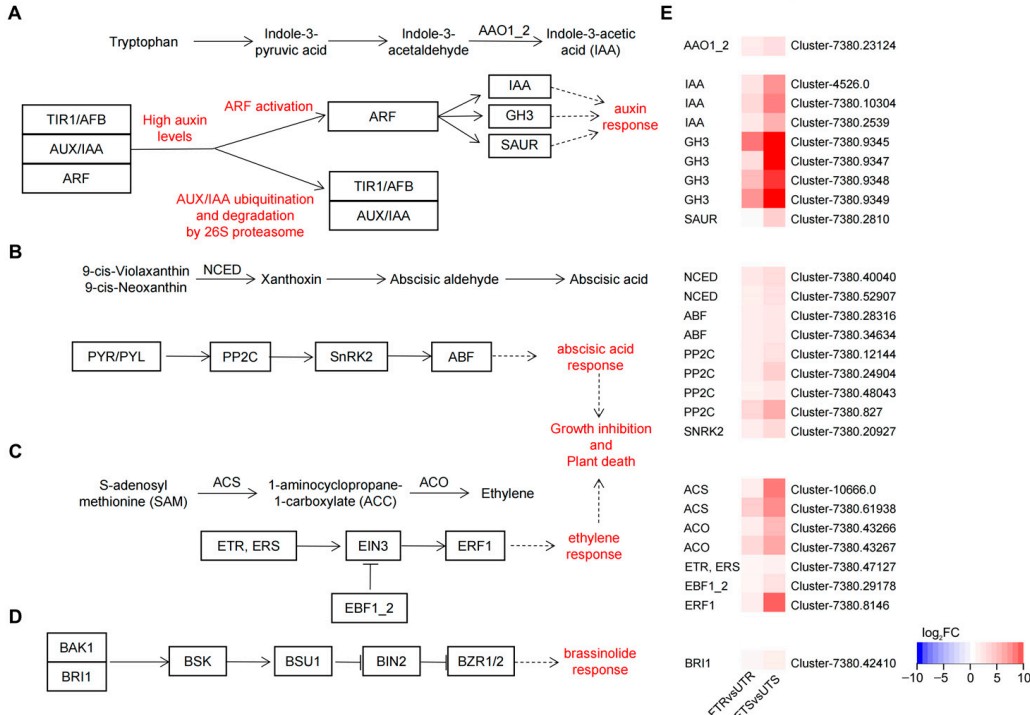

**Figure 6.** Biosynthesis and signal transduction pathways of IAA, ABA, ETH, and BR, and the expression levels of related genes in the S and R comparison settings. (**A**) IAA biosynthesis and signal transduction pathways. (**B**) ABA biosynthesis and signal transduction pathways. (**C**) ETH biosynthesis and signal transduction pathways. (**D**) BR signal transduction pathways. (**E**) Expression levels of related genes in the S and R comparison settings. Arrows indicate direct (solid line) and indirect (dotted line) activation, hammer-ended lines indicate inhibition; black words within a solid black line box represent gene products (mostly components of the plant hormone signal transduction pathway but including enzymes); black words without a box represent substrates or products; and red words without a box represent biological responses or biological processes.

Auxin regulates ethylene biosynthesis through the auxin-induced expression of 1-aminocyclopropane-1-carboxylate synthase (ACS), which catalyzes the rate-limiting step of ethylene synthesis [38–40]. The induction of increased expression of genes encoding ACS by SAHs has also been reported several times [41–43]. Our transcriptomic data are consistent with these reported findings, and the results suggested that after 24 h of florpyrauxifen-benzyl treatment, two genes encoding ACS (Cluster-10666.0; Cluster-7380.61938) were upregulated in both FTR and FTS compared with the untreated control (UTR and UTS), but the fold change (FC) in gene expression of these two genes encoding ACS was significantly higher in S than in the R biotype. This suggests that the S biotype may have higher ethylene biosynthetic activity in response to florpyrauxifen-benzyl treatment, which is also consistent with the upregulated expression of the ethylene receptor and ethylene response sensor (ETR, ERS) gene, ethylene-responsive transcription factor 1 (ERF1), and the upregulated expression of EIN3-binding F-box protein (EBF, which is ubiquitinated to degrade EIN3) in the ETH signaling pathway of FTS observed in our transcriptome data. The increased ethylene synthesis induced by florpyrauxifen-benzyl as an SAH and the active ETH signaling pathway caused by ethylene accumulation may be the key to accelerating the death of S barnyard grass [44].

SAHs also upregulate the expression of the gene encoding 9-cis-epoxy carotenoid dioxygenase (NCED), a key enzyme in ABA biosynthesis [41,45]. Transcriptome analysis showed that two genes encoding 9-cis-epoxy carotenoid dioxygenase (NCED) were upregulated in both R and S biotypes after 24 h of florpyrauxifen-benzyl treatment compared with the untreated control (Cluster-7380.40040; Cluster-7380.52907). However, the fold change (FC) in gene expression of these two genes was significantly higher in the S than in the R biotype. This suggests that florpyrauxifen-benzyl induces higher expression of related genes involved in ABA biosynthesis in S barnyard grass, which is consistent with the observation from our transcriptomic data that the SNRK2 gene (Cluster-7380.20927) and protein phosphatase 2C (PP2C) gene in the ABA signaling pathway were more upregulated in the S biotype after florpyrauxifen-benzyl treatment than in R. The increased ABA synthesis induced by florpyrauxifen-benzyl as an SAH and the active ABA signaling pathway caused by ABA accumulation may also be the key to accelerating the death of S barnyard grass [44].

The transcriptome results showed that the same gene encoding the gibberellin receptor (GID1) was upregulated in both R and S biotypes after florpyrauxifen-benzyl treatment (Cluster-7380.24813), but the differential expression was similar in R and S barnyard grasses (Supplementary Table S6). This suggests that although florpyrauxifen-benzyl can induce activation of the GA signaling pathway in barnyard grass, it is possible that the GA signaling pathway may not be associated with the different tolerances of R and S barnyard grass to florpyrauxifen-benzyl.

BR and IAA exert synergistic effects on the regulation of plant growth and development. Both BR and IAA promote cell expansion and some BR-induced genes are upregulated by IAA [46]. The synergistic effects of BR and IAA are thought to be associated with the activity of some ARFs [47]. In addition, BR promotes the biosynthesis and signaling pathways of ETH and ABA at high concentrations and accelerates the early stages of leaf senescence [48]. Our transcriptome results showed that florpyrauxifen-benzyl treatment upregulated the expression of genes related to the BR signaling pathway, and this upregulation was more pronounced in S than in R barnyard grass (one BR-insensitive 1 (BRI1) was upregulated in S (Cluster-7380.42410). This may be a reason for the more-intense ETH and ABA responses in the S biotype after florpyrauxifen-benzyl treatment compared with those in R, and it accelerates the leaf senescence of S barnyard grass.

## 5. Conclusions

Overall, our study is the first transcriptome analysis of the differential effects of florpyrauxifen-benzyl treatment between florpyrauxifen-benzyl-resistant and -susceptible *E. crus-galli*. This study reflected the differences in phytohormone biosynthesis and signal

transduction between R and S barnyard grasses in response to florpyrauxifen-benzyl treatment and will be helpful for understanding the phytotoxicity mechanisms of florpyrauxifen-benzyl. The DEGs discussed in this study were mainly associated with phytohormone biosynthesis and signal transduction, and the results are consistent with and provide transcriptomic evidence for previously reported changes in phytohormone components induced by florpyrauxifen-benzyl or other synthetic auxin herbicides [19,41–43,45]. However, this study has two limitations. First, our study revealed differences in phytohormone biosynthesis and signal transduction between R and S barnyard grasses in response to florpyrauxifen-benzyl treatment; however, we did not clarify the reasons for these differences. Second, only one resistant and one sensitive biotype of barnyard grass variant were used for transcriptome analysis, and the diversity of biotypes was insufficient; therefore, the conclusions may not be generalizable.

**Supplementary Materials:** The following supporting information can be downloaded at https://www.mdpi.com/article/10.3390/agronomy13030702/s1: Supplemental Table S1. Overview of the sequencing reads obtained from each sample; Supplemental Table S2. Functional annotations of unigenes in the databases; Supplemental Table S3. Go function enrichment of up- or down-regulated genes at untreated compared with florpyrauxifen-benzyl treated in R; Supplemental Table S4. Go function enrichment of up- or down-regulated genes at untreated compared with florpyrauxifen-benzyl treated in S; Supplemental Table S5. The sequences of RT-PCR primers; Supplemental Table S6. The hormone-related DEGs after treatment with florpyrauxifen-benzyl in both R and S biotypes.

**Author Contributions:** Conceptualization, J.Z. and Y.L.; data curation, W.J. and J.S.; formal analysis, W.J. and Y.Y.; funding acquisition, W.T. and Y.L.; investigation, J.S. and J.Z.; methodology, Y.Y. and J.Z.; project administration, W.T. and X.Y.; resources, W.T. and Y.Y.; software, J.S. and X.Y.; supervision, W.T. and Y.L.; validation, J.S. and X.Y.; visualization, W.J. and X.Y.; writing—original draft, W.J. and X.Y.; writing—review and editing, W.J. and X.Y. All authors have read and agreed to the published version of the manuscript.

**Funding:** This work was financially supported by the China Agriculture Research System (CARS-01-02A), the Open Project Program (20210304) of State Key Laboratory of Rice Biology and Breeding and the Rice Pest Management Research Group of the Agricultural Science and Technology Innovation Program, Chinese Academy of Agricultural Sciences.

**Institutional Review Board Statement:** Not applicable.

**Informed Consent Statement:** Not applicable.

**Data Availability Statement:** The raw reads have been deposited in the NCBI Sequence Read Archive (SRA) database (BioProject PRJNA938437). All other data generated or analyzed during this study are included in this article and its supplementary information files.

**Acknowledgments:** We would like to thank all our colleagues in the laboratory who managed the laboratory and took care of the plant material used for this study, ensuring the smooth running of this research.

**Conflicts of Interest:** The authors declare no conflict of interest.

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
