# Peer review of "Comparative Transcriptome Analysis of the Differential Effects of Florpyrauxifen-Benzyl Treatment on Phytohormone Transduction between Florpyrauxifen-Benzyl-Resistant and -Susceptible Barnyard Grasses (Echinochloa crus-galli (L.) P. Beauv)"

_agronomy, doi:10.3390/agronomy13030702_

Round 1

Reviewer 1 Report

Review of ms. agronomy-2185511 entitled Comparative Transcriptome Analysis of the Differential Effects of Florpyrauxifen-benzyl Treatment on Phytohormone Transduction between Florpyrauxifen-benzyl-resistant and Susceptible Barnyard grasses(Echinochloa crus-galli (L.) P.Beauv)

Summary

The paper aims to understand the phytotoxicity and resistance mechanisms of Echinocloa Crus-galli caused by the florpyrauxifen-benzyl herbicide. For this purpose, the authors applied RNA-seq gene expression profiling of treated and untreated sensitive (S) and resistant (R) biotypes of E. crus-galli. They found upregulation of genes involving phytohormone signaling, auxin response, ethylene, and abscisic acid biosynthesis in the S against R biotypes caused by 24h after the treatment. Higher gene expression of genes taking part in BR pathway was also found in S biotype.

The manuscript is clear, relevant to the field, and well-structured however, it needs more precise elaboration in several paragraphs, with special regard to the methodological part and discussion. The cited references are relevant. The chosen method is suitable for supporting the hypothesis, although the concept that the article seeks to answer (hypothesis) is not explained anywhere in the text. The conclusion should be worded more nuanced since gene expression was examined, and physiological and analytical data do not support the statements.

After major revising the manuscript, I recommend it is acceptable for publication.

Comments and suggestions for Authors:

Abstract

-   Line 14-19. Please reword the sentence “To understand….” more concise, so it is too long, or take it in two sentences.

-   Since authors found differences in the genes of the biosynthetic pathways, they cannot say that more active biosynthesis, etc. would require an analytical study of gene products. Please reword it for example using the expression “genes involved in biosynthesis, signal transduction, etc.)

-   In the abstract, please don’t abbreviate BR (brassinosteroid).

Introduction

-   The authors should describe at the end of this paragraph what the hypothesis is, the purpose of the study, and what they would like to answer (based on the literature, that is a shortcoming in the field, etc.). Also, how do the given method and test give more than before?

-   Please summarize and explain with references that the applied methods are appropriate and can be successfully used to solve the problem

Materials and Methods

-   The authors should describe the low and high doses more precisely in this section. Address what the authorized doses are in the license document and how much it is calculated in laboratory conditions.

-   Please explain why the sample was collected exactly 24 hours after treatment (which seems good). Is this based on a preliminary experiment or according to a standard protocol?

-   Processing of bioinformatics should be explained in much more detail, including the programs and parameters used.

Results

-   Why were the uniquely expressed genes not filtered out in the given samples? And the genes expressed only in the treated samples? This can be determined based on the transcript abundance table and would be informative.

-   Are the genes selected for q-PCR among the top 100 genes expressed in the treated samples?

-   Figure 2. Should be sharper. I think a heat map would be interesting.

Discussion

-   Please explain in more detail the results and stated conclusions with references.

-   I think the expression “genetic variation” is not correct because it refers more to snp or mutations in genes. Please reword these sentences.

-   Please relate also the results of the KEGG pathway analysis to the genes discussed.

-   Based on the gene expression data the authors guess that ABA and ETH accumulation accelerates the death of the plant, the statements aimed at this should be formulated in a more nuanced way since this should also be proven analytical experiments. If no analytical analysis is possible, please support these phenomena by the scientific background (literature).

-   line 363. Please reword the expression “We believe that…” and replace more scientific wording.

-   I think should be informative a figure summarizing the main points of results (e.g. ETH, ABA, BR pathway genes, and signaling as a response to the treatment).

Conclusion

-   Please also address the significance of the results and the limitations of the study presented.

Author Response

Dear Professor:

Thank you for your review comments on our manuscript entitled "Comparative Transcriptome Analysis of the Differential Effects of Florpyrauxifen-benzyl Treatment on Phytohormone Transduction between Florpyrauxifen-benzyl-resistant and Susceptible Barnyard grasses (Echinochloa crus-galli (L.) P.Beauv)". (ID: agronomy-2185511). These comments are all valuable and very helpful in revising and improving our paper and of great importance to our research. We have carefully studied the comments and made corrections that we hope will meet with approval. The revised parts are indicated in the paper by colored words or colored background. The major corrections in the paper and the responses to the reviewers' comments are as follows:

Responds to the reviewe comments:

Point 1: To reword the sentence “To understand….” more concise

Response 1: Thank you very much for your correction. Based on your and other reviewers' suggestions, we have revised this section. The revised content no longer addresses herbicide resistance, which is more appropriate to the content and title of the article, and the content has become more concise. (line 15-19)

Point 2: To reword “more active biosynthesis” for example using the expression “genes involved in biosynthesis, signal transduction, etc.)

Response2: Thank you very much for your correction. We have revised the expression "more active biosynthesis" to "higher expression of relevant genes involved in the synthesis/transduction pathway" as you suggested.(line 24, line 408, etc.)

Point 3: Don’t abbreviate BR (brassinosteroid).

Response 3: Based on your suggestion, we have changed the abbreviation “BR” to the full name “brassinosteroid”.(line 25)

Point 4: Describe at the end of the "Introduction" section of the hypothesis, the purpose of the study.

Response 4: Thank you for your professional advice, we have rewritten the last paragraph of the "Introduction" section to clarify our hypothesis and the purpose of the study. (line 101-117)

Point 5: To summarize and explain with references that the applied methods are appropriate and can be successfully used to solve the problem

Response 5: Thank you for your expert advice, we have briefly described the current status of transcriptomics applications in the area of plant-herbicide interactions and cited references to prove our point. This is a well-established technique that can help us complete our study. (line 103-110)

Point 6: Describe the low and high doses more precisely in this section. Address the authorized doses.

Response 6: Based on your suggestions we have added the recommended doses of florpyrauxifen-benzyl in the field and used a table to list the doses we used to make the dose-response curves in this experiment. (line 158-168)

Point 7: Explain why the sample was collected exactly 24 hours after treatment.

Response 7: Because this study is the first transcriptome study on clopyralid and barnyard grass, we consulted some previous studies on herbicides with transcriptome technology and found that they chose 24 h [DOI: 10.1016/j.pestbp.2019.04.017], meanwhile, there are also some studies on other growth hormone herbicides that show relatively significant hormonal changes in plants after 24 h of exposure, so we chose to use a 24 h post-dose sample to measure the transcriptome [DOI: 10.1016/j.pestbp.2021.105007].

Point 8: To explain the processes of bioinformatics in more detail.

Response 8: We have rewritten this section to describe the R package used and the parameters set. (line 202-220)

Point 9: Why were the uniquely expressed genes not filtered out in the given samples? And the genes expressed only in the treated samples?

Response 9: Genes marked as FALSE in the supplemental Table are not uniquely expressed genes but may be one or more of the P values, Q values, or FC in the transcriptome sequencing results that do not match the conditions we set.  For the genes involved in plant hormone signaling and labeled as FALSE in the study, we performed qPCR to verify that they were expressed in both treated and untreated samples.

Point 10: Are the genes selected for q-PCR among the top 100 genes expressed in the treated samples?

Response 10: Thank you for your suggestion, but the top 100 genes expressed were not a condition for our selection of genes for qPCR validation. We selected genes involved in plant hormone signaling related to the title and discussion of the article for qPCR validation, and we did not focus on whether they belonged to the top 100 genes expressed.

Point 11: Suggest changing Figure 2 to a heat map.

Response 11: Thank you for your suggestion, I tried but failed. The TOP20 pathways enriched with the genes in the R and S biotype comparison setups are similar but not identical, and the key element we want to discuss, phytohormone signaling, becomes less prominent when we use the heatmap, so we decided to keep that part.

Point 12: Explain in more detail the results and stated conclusions with references.

Response 12: We have added to the discussion section based on your next suggestions in points 17 and 18.

Point 13: The expression “genetic variation” is not correct

Response 13: Thank you for your correction, the original "the fold of genetic variation" has been replaced with "the Fold-Change(FC) in gene expression". (line 391 and 407)

Point 14: Relate also the results of the KEGG pathway analysis to the genes discussed.

Response 14: Based on your suggestion we added a discussion of the KEGG enrichment results and explained our reasons for focusing on the phytohormone signaling pathway. (line 340-351)

Point 15: Based on the gene expression data the authors guess that ABA and ETH accumulation accelerates the death of the plant, the statements aimed at this should be formulated in a more nuanced way since this should also be proven analytical experiments. If no analytical analysis is possible, please support these phenomena by the scientific background (literature).

Response 15:The accumulation of ABA and ETH accelerates the response of plants to dieback, probably by promoting the production of reactive oxygen species that ultimately lead to plant death, which has been described in the toxicological mechanism of quinclorac. Because quinclorac and florpyrauxifen-benzyl belong to the same HRAC group O, we made this inference based on the results of quinclorac studies reported by previous authors. We have added references in the text. (line 401 and 415)

Point 16: line 363. Please reword the expression “We believe that…” and replace more scientific wording.

Response 16: Thanks for your correction, we have modified the expression "We believe that..." and the expression has become more scientific.

Point 17: Should be informative a figure summarizing the main points of results (e.g. ETH, ABA, BR pathway genes, and signaling as a response to the treatment).

Response 17: Thanks to your suggestion, we have created Figure 5 to show the main results of the discussion section.

Point 18: Please also address the significance of the results and the limitations of the study presented.

Response 18: Thank you for your professional advice. We have rewritten this section based on your suggestions, and the rewritten content shows the importance and limitations of this study.

Special thanks to you for your good comments.

We tried our best to improve the manuscript and made some changes in the manuscript. These changes will not influence the content and framework of the paper. And here we did not list all the changes but marked in colored words or colored background in revised paper.

We appreciate for your warm work earnestly, and hope that the correction will meet with approval.

Once again, thank you very much for your comments and suggestions.

Reviewer 2 Report

It should be of interest to indicate to which HRAC groups the herbicides in which resistance has been described in rice belong, not only in China but in the Word. In this way, the resistance found to Florpyrauxifen-benzil in Echinochloa would be better framed. For example, quinclorac mentioned in the introduction belongs to the same HRAC-group than Florpyrauxifen-benzil. This is important for the point of view of managing herbicides in the field (rotation of a.i.) to avoid the development of resistance. 

Author Response

Dear Professor:

Thank you for your review comments on our manuscript entitled "Comparative Transcriptome Analysis of the Differential Effects of Florpyrauxifen-benzyl Treatment on Phytohormone Transduction between Florpyrauxifen-benzyl-resistant and Susceptible Barnyard grasses (Echinochloa crus-galli (L.) P.Beauv)". (ID: agronomy-2185511). These comments are all valuable and very helpful in revising and improving our paper and of great importance to our research. We have carefully studied the comments and made corrections that we hope will meet with approval. The revised parts are indicated in the paper by colored words or colored background. The major corrections in the paper and the responses to the reviewers' comments are as follows:

Responds to the reviewe comments:

Point 1: To indicate to which HRAC groups the herbicides in which resistance has been described in rice belong, not only in China but in the Word.

Response 1: Thank you for your professional advice, which has given us new ideas. We have revised the "Introduction" section to describe the global status of herbicide resistance in rice barnyard grass from the perspective of the HRAC groups and to provide examples of the major herbicides that have developed resistance within each HRAC group.

Special thanks to you for your good comments.

We tried our best to improve the manuscript and made some changes in the manuscript. These changes will not influence the content and framework of the paper. And here we did not list all the changes but marked in colored words or colored background in revised paper.

We appreciate for your warm work earnestly, and hope that the correction will meet with approval.

Once again, thank you very much for your comments and suggestions.

Reviewer 3 Report

Dear authors,

The study of florpyrauxifen-benzyl effect and resistance in weeds has merit, because it is a new auxinic herbicide, with recent resistance reports and still not well elucidated. This study presents the first transcriptome analysis of this herbicide, but the analysis chosen was not adequate to study the resistance mechanism. The study cannot focus on herbicide resistance genes identification because the contrasts analyzed in the manuscript cannot answer this question. The correct contrasts should be R treated x S treated and R untreated x S untreated, to identify genes that are exclusively up-regulated in R. However, the contrasts chosen by the authors are suitable to understand the genes/pathways affected by the auxinic herbicide florpyrauxifen-benzyl. In this scenario, the authors have two options 1) do not focus on herbicide resistance and keep the analysis focusing on genes/paths affected by florpyrauxifen-benzyl, or 2) reanalyze the data contrasting R treated x S treated and R untreated x S untreated, and then you can explore the herbicide resistance candidate genes. In both cases, the authors should rewrite the objectives, results and discussion session, to be more clear.

The Material and Methods session is not completed, and the description of the experiments is not well organized and missing essential information. For example, there is no information about florpyrauxifen-benzyl doses used to perform dose-response curves in the Material and Methods session. Was the barnyardgrass soil kept flooding? What was the variable evaluated? How many days after treatment? We can find some of this information in the Results session, but it must be in the Material and Methods session.

Some interpretations are not appropriate. For example: “This suggests that florpyrauxifen-benzyl induces more ethylene synthesis in S…” Is this a cause or a consequence? The authors consider it a cause. However, the R biotype produces less ethylene because it can survive the herbicide application, probably detoxifying the herbicide. The resistance causes are not investigated in this study. The authors have data to do it, but the contrasts were not adequate.

The whole manuscript has not adequate paragraph construction. A paragraph should be composed of two or more sentences. For example, line 137 and many others. One sentence should not be an individual paragraph. Another example is the paragraph from line 296, it should be together with the previous one. The same for the paragraph from line 316.

I would like to reinforce, the authors have the data, they just need to explore it in a different way. After doing it, the authors can re-submit the manuscript to Agronomy.

Other considerations:

Abstract and Introduction:

Line 27 = E. crus-galli should be italic.

Line 42-49: Put together paragraphs two and three.

Paragraph four: Remove the capital letters from the entire word. Example: TRANSPORT INHIBITOR RESISTANT 1/AUXIN SIGNALING F-BOX à Transport Inhibitor Resistant 1/Auxin Signaling F-Box.

Line 65: Is quinclorac an auxinic herbicide when acting in grasses? It has another mode of action suggested for grasses , even if it is not already 100% elucidated.

Line 66: Start a new paragraph “Recently, florpyrauxifen-benzyl…”

Line 69 = E. oryzicola should be italic.

Line 74: replace “may eventually lead to the…” for “already selected…” because the barnyardgrass resistance to florpyrauxifen is already reported.

 Material and Methods:

Line 107: Was the R biotype collected in the field after one season of florpyrauxifen-benzyl spray? Because the herbicide was released in China in 2017, and the biotype was also collected in 2017.

Line 109: Replace F2 by G2, because F (F1, F2,..) is used to refer to hybrid. For example, crossing R x S biotype we can use F1. The most appropriate to refer to generations is G.

Line 109: Keep this paragraph together with the previous one.

All experiments to evaluate florpyrauxifen-benzyl resistance were conducted from May to July 2020 and validated with three replications in 2021 and 2022 (at specific concentrations of florpyrauxifen-benzyl…” Not clear. Which experiments were performed from May to July 2020? Dose-response curves? What do you mean with validation in 2021 and 2022? How many times were the dose-response curves repeated? How were the different dose-response curves runs analyzed? Explain better.

Pay attention to non-necessary capital letters in the manuscript. For example “…for evaluation of Florpyrauxifen-benzyl…”

Were the tissue samples sent to Novogene to extract RNA? Which protocol did they use to extract RNA? The sentence “RNA extraction and RNA-seq were performed using Novogene (Beijing, China)” is not clear.

Line 181: “Actin was used as a housekeeping gene”. How many others housekeeping genes were evaluated? Do you have any reference to choose actin?

 Results

 The Table 1 information must be shown as a Figure. An appropriate Table 1 should have the curve parameters, GR50 and RI, and their respective significance. 

Why have you chosen de novo assembly? The Echinochloa crus-galli genome (version 3) is available: “Genomic insights into the evolution of Echinochloa species as weed and orphan crop”.

Author Response

Dear Professor:

Thank you for your review comments on our manuscript entitled "Comparative Transcriptome Analysis of the Differential Effects of Florpyrauxifen-benzyl Treatment on Phytohormone Transduction between Florpyrauxifen-benzyl-resistant and Susceptible Barnyard grasses (Echinochloa crus-galli (L.) P.Beauv)". (ID: agronomy-2185511). These comments are all valuable and very helpful in revising and improving our paper and of great importance to our research.

For the two options in your comment, we finally decided to go with the first one, which is do not focus on herbicide resistance and keep the analysis focusing on genes/paths affected by florpyrauxifen-benzyl. All changes in this paper are based on this choice, and we have removed the discussion of resistance mechanisms because our study is not very relevant to resistance. This study focus on the difference in phytohormone biosynthesis and signal transduction between R and S barnyard grasses in response to florpyrauxifen-benzyl treatment but without clarifying the reasons for these differences,which is the limitation of this study. We have carefully studied the comments and made corrections that we hope will meet with approval.

The revised parts are indicated in the paper by colored words or colored background. The major corrections in the paper and the responses to the reviewers' comments are as follows:

Responds to the reviewe comments:

Point 1: Line 27 = E. crus-galli should be italic.

Response 1: Thank you for the correction. We have noted the error you pointed out and have corrected it and will try to avoid similar errors in the future. (line 28)

Point 2: Line 42-49: Put together paragraphs two and three.

Response 2: Thank you for the correction. We have noted the error you pointed out and have corrected it and will try to avoid similar errors in the future. (line 47-54)

Point 3:Paragraph four: Remove the capital letters from the entire word. Example: TRANSPORT INHIBITOR RESISTANT 1/AUXIN SIGNALING F-BOX à Transport Inhibitor Resistant 1/Auxin Signaling F-Box.

Response 3: Thank you for the correction. We have noted the error you pointed out and have corrected it and will try to avoid similar errors in the future. (line 57 and 58)

Point 4: Line 65: Is quinclorac an auxinic herbicide when acting in grasses? It has another mode of action suggested for grasses , even if it is not already 100% elucidated.

Response 4: This is an interesting question, although according to data from The International Herbicide-Resistant Weed Database and related descriptions, there is a resistance to the mechanism of action in barnyard grass in HRAC group L (quinclorac with the mechanism of action in monocots), there is also more resistance in HRAC group O (quinclorac). Also, in many studies related to the resistance of barnyard grass to quinclorac, it is still described as a synthetic auxin herbicide. As for the mechanism of action (MOA) in monocots, that is, the mechanism involving cyanide accumulation and β-CAS detoxification, as you said, "it is not already 100% elucidated". And according to a study by Satoshi Iwakami's team in 2019, "Quinclorac resistance in Echinochloa phyllopogon is associated with reduced ethylene synthesis rather than enhanced cyanide detoxification by β-cyanoalanine synthase", there are some unknown parts of this mechanism. However, in general, there are still many resistant barnyard grasses that are resistant to quinclorac of HRAC group O and not to quinclorac of HRAC group L (quinclorac MOA in monocots), and the quinclorac resistance described in this paper is an objective fact.

Point 5: Line 66: Start a new paragraph “Recently, florpyrauxifen-benzyl…”

Response 5: Thank you for the correction. We have noted the error you pointed out and have corrected it and will try to avoid similar errors in the future. (line 72)

Point 6: Line 69 = E. oryzicola should be italic.

Response 6: Thank you for the correction. We have noted the error you pointed out and have corrected it and will try to avoid similar errors in the future. (line 75)

Point 7: replace “may eventually lead to the…” for “already selected…” because the barnyardgrass resistance to florpyrauxifen is already reported.

Response 7: Thank you for the correction. We have noted the error you pointed out and have corrected it and will try to avoid similar errors in the future. (line 80)

Point 8: Line 107: Was the R biotype collected in the field after one season of florpyrauxifen-benzyl spray? Because the herbicide was released in China in 2017, and the biotype was also collected in 2017.

Response 8: We are sorry that we do not have information about whether the R biotype has a history of treatment with florpyrauxifen-benzyl. The seed of biotype R is one of the barnyard grass seeds collected by our team from all over China. After the release of clopyralid in China we selected a large number of barnyard grass populations from our seed bank to test the efficacy of florpyrauxifen-benzyl, and by chance, we found that biotype R is a florpyrauxifen-benzyl-resistant biotype, so we conducted the follow-up study on it.

Point 9: Line 109: Replace F2 by G2, because F (F1, F2,..) is used to refer to hybrid. For example, crossing R x S biotype we can use F1. The most appropriate to refer to generations is G.

Response 9: Thank you for your professional advice, we have adopted and modified the relevant content according to your suggestions. (line 123 and 146)

Point 10: Line 109: Keep this paragraph together with the previous one.

Response10: Thank you for the correction. We have noted the error you pointed out and have corrected it and will try to avoid similar errors in the future. (line 124) 

Point 11: “All experiments to evaluate florpyrauxifen-benzyl resistance were conducted from May to July 2020 and validated with three replications in 2021 and 2022 (at specific concentrations of florpyrauxifen-benzyl…” Not clear.

Response 11: Thank you very much for your professional opinion and we have rewritten this section. It is now presented more clearly and accurately. The data used to create the dose-response curves in this paper are the results of trials conducted in 2020. Each trial had three replicates, and the trials were conducted at least twice to ensure reliability.

Point 12: Pay attention to non-necessary capital letters in the manuscript. For example “…for evaluation of Florpyrauxifen-benzyl…”

Response 12: Thank you for your correction. Due to the rewriting of this paragraph, we changed the title to "Dose-response experiments of florpyrauxifen-benzyl", and we have noted the error you pointed out and tried to avoid similar errors. (line 128)

Point 13: The sentence “RNA extraction and RNA-seq were performed using Novogene (Beijing, China)” is not clear.

Response 13: Thank you very much for your correction. We have rewritten this part, and the rewritten content explains that Novogene extracted RNA from the leaf tissue samples that we sent there, and also explains the kit that was used for RNA extraction. (line 184-199)

Point 14: Line 181: “Actin was used as a housekeeping gene”. How many others housekeeping genes were evaluated? Do you have any reference to choose actin?

Response 14: I apologize for my inaccurate terminology that caused you to misunderstand. We screened several housekeeping genes and finally decided to use actin as the only housekeeping gene in this study. Based on your suggestion, we have changed the expression "a housekeeping gene" to "the housekeeping gene" and added the reference. (line 232)

Point 15: The Table 1 information must be shown as a Figure. An appropriate Table 1 should have the curve parameters, GR50 and RI, and their respective significance.

Response 15: Thank you for your professional advice. We have taken your suggestion and used a different calculation method than before to create a dose-response curve from the data in the original Table 1, and replaced the original Table 1 with Figure 1.

Point 16: Why have you chosen de novo assembly? The Echinochloa crus-galli genome (version 3) is available: “Genomic insights into the evolution of Echinochloa species as weed and orphan crop”.

Response 16: Thank you for your correction, our transcriptome was made in November 2021, and the article was published in February 2022. The article was not published when we delivered Novogene to sequence the transcriptome, so we chosen de novo assembly at that time.

Special thanks to you for your good comments.

We tried our best to improve the manuscript and made some changes in the manuscript. These changes will not influence the content and framework of the paper. And here we did not list all the changes but marked in colored words or colored background in revised paper.

We appreciate for your warm work earnestly, and hope that the correction will meet with approval.

Once again, thank you very much for your comments and suggestions.

Round 2

Reviewer 1 Report

Dear Authors,

Thank you for your responses and corrections of the ms.

I accept and recommend the manuscript for publication with the perfomed corrections.

Author Response

Response to Reviewer 1 Comments

Dear Professor:

Thank you for your acceptance and recommendation of our manuscript entitled "Comparative Transcriptome Analysis of the Differential Effects of Florpyrauxifen-benzyl Treatment on Phytohormone Transduction between Florpyrauxifen-benzyl-resistant and Susceptible Barnyard grasses (Echinochloa crus-galli (L.) P.Beauv)". (ID: agronomy-2185511).

In this round of manuscript revision, we have again improved the manuscript with the comments of other reviewers, and also polished up the research design, methods, results and conclusions of the manuscript. Overall, the manuscript has been improved from the previous version.

Special thanks to you for your good comments.

We tried our best to improve the manuscript and made some changes in the manuscript. These changes will not influence the content and framework of the paper. And here we did not list all the changes but marked in colored words or colored background in revised paper.

We appreciate for your warm work earnestly, and hope that the correction will meet with approval.

Once again, thank you very much for your comments and suggestions.

Reviewer 3 Report

Dear authors,

The manuscript improved after the first revision, mainly regarding the previous wrong data discussion based on wrong RNAseq contrasts. However, the manuscript still has some problems, and it is now suitable to be published in a journal impact factor 3.9. The material and methods session still has problems. The discussion session is still poor and almost does not discuss the results. The conclusion also should be improved. The data presented are important to understand the florpyrauxifen-benzyl effect on hormone pathways, but it should be better discussed, and unfortunately, the manuscript structure is not adequate to be published. I recommend improving the structure quality and publishing it in another journal.  

Line 76: remove the word “However”.

Line 77: remove the word “urgent”.

Line 76-77: rewrite the sentence. Florpyrauxifen-benzyl is repeated twice and it does not sound good.

Line 91: replace the sentence by “Furthermore, the evolution of florpyrauxifen-benzyl resistance in multiple-resistant barnyard grass can be linked to non-target-site resistance mechanisms…”

Line 136: “The experiment was repeated three times”; Line 163: “repeated at least twice”. How many times was the experiment repeated?

Line 139: Session “Seedling preparation for transcriptome analysis”: what tissue did you collect? Young or old leaves?

Line 152: Please, double-check is the label doses 18 to 36 g a.i. ha−1 are correct.

Line 155: Remove the sentence “When the herbicide was applied, the low dose was applied first, followed by the higher dose”.

Line 157: The doses from Table 1 should be in the text of the “Dose-response experiments of florpyrauxifen-benzyl” session. It is not necessary a table to show the doses.

Line 174: E. crus-galli should be italic

Line 177: “UTR_1, UTR_2, UTR_3, UTS_1, UTS_2, UTS_3, FTR_1, FTR_2, FTR_3, 177 FTS_1, FTS_2, and FTS_3” = It is not necessary to show all replicate names. You just need to show the treatment name once.

Line 190: replace “Difffferential” by Differential

Line 190 - Session “Differential Gene Expression and Enrichment Analysis”: describe better the RNAseq analysis. Which software did you use to treat the data? Which software did you use to align the reads? To build the transcriptome? Which software did you use to blast with NR database? What Log2FC did you use as threshold?

Line 192: DEGseq2? Or DEseq2?

Line 216 and 221: Replace qRT–PCR for RT-qPCR

Paragraph line 240 and paragraph line 243: you should build one paragraph. A paragraph cannot be only one sentence. The same problem for paragraphs from lines 256 and 259.

Line 290-295: Add space between the word and the (, for example: “ABA(abscisic acid)”

Line 301: The paragraph cannot be only one sentence.

Line 301: “Thirty-two” remove the capital letter.

Line 302 and 307: “Quantitative Real-Time PCR (qRT– PCR)” is wrong, the quantitative is the PCR, then the correct is RT-qPCR.

Line 344-345: “The reason for the more intense auxin response in S may be related to IAA biosynthesis”: I think the main reason is because the R biotype degrades the herbicide faster, and avoid the auxin cascade effect.

Line 372 -376: this is results, not discussion.

Line 399: You discuss a new gene GID1, that was not shown before.

Line 432-437: The conclusion cannot finish like this.

Figure 1: The equation and parameters are too small.

Figure 3: too small as well.

Figure 4: What is x axis?  and y axis?

Table 2: How and why did you choose these genes to analyze in qPCR? What about the other ~400 genes up- and down-regulated in both contrasts?

Figure 5: The genes shown in the heat map do not match the results shown in Table 2. For example, ACS is up-regulated in both contrasts, but in Figure 5 it is down-regulated for R contrast. ACO as well.

Author Response

Response to Reviewer 3 Comments

Dear Professor:

Thank you for your review comments on our manuscript entitled "Comparative Transcriptome Analysis of the Differential Effects of Florpyrauxifen-benzyl Treatment on Phytohormone Transduction between Florpyrauxifen-benzyl-resistant and Susceptible Barnyard grasses (Echinochloa crus-galli (L.) P.Beauv)". (ID: agronomy-2185511). These comments are all valuable and very helpful in revising and improving our paper and of great importance to our research.

We have carefully studied the comments and made corrections that we hope will meet with approval.

The revised parts are indicated in the paper by colored words or colored background. The major corrections in the paper and the responses to the reviewers' comments are as follows:

Responds to the reviewe comments:

Point 1: Line 27 = E. crus-galli should be italic.

Response 1: Thank you for the correction. But this error we have solved in the previous version of the revised manuscript ( agronomy-2185511_V2 ). We have again carefully checked all the scientific names in the full text in this version of the manuscript to ensure that there are no similar errors.

Point 2: Line 76: remove the word “However”.

Response 2: Thank you for the correction. We have noted the error you pointed out and have corrected it. (line 78)

Point 3: Line 77: remove the word “urgent”.

Response 3: Thank you for the correction. We have noted the error you pointed out and have corrected it. (line 79)

Point 4: Line 76-77: rewrite the sentence. Florpyrauxifen-benzyl is repeated twice and it does not sound good.

Response 4: Thank you for the correction. We replaced the second "florpyrauxifen-benzyl" by "it". (line 78-79)

Point 5: Line 91: replace the sentence by “Furthermore, the evolution of florpyrauxifen-benzyl resistance in multiple-resistant barnyard grass can be linked to non-target-site resistance mechanisms…”

Response 5: Thank you for your suggestion, we changed the sentence as you suggested and now it looks much better. (line 95-98)

Point 6: Line 136: “The experiment was repeated three times”; Line 163: “repeated at least twice”. How many times was the experiment repeated?

Response 6: Thank you for the correction. The experiment was repeated three times. We unified this description to avoid ambiguity. (line 141 and 177)

Point 7: Line 139: Session “Seedling preparation for transcriptome analysis”: what tissue did you collect? Young or old leaves?

Response 7: Thank you for the correction, it is the old leaves, we have replaced the original "samples" to "old leaves". (line 159)

Point 8: Line 152: Please, double-check is the label doses 18 to 36 g a.i. ha−1 are correct.

Response 8: Thanks to your kind reminder, we determined that this label dose is correct. From the China Pesticide Information Network, we can check the recommended field dose of 3% Rinskor for 40 to 80 ml per Chinese mu, one hectare equals 15 Chinese mu, which translates into an active ingredient content of 18 to 36 g a.i. ha−1.

Point 9: Line 155: Remove the sentence “When the herbicide was applied, the low dose was applied first, followed by the higher dose”.

Response 9: Thank you for your suggestion and based on your suggestion we have removed this sentence.

Point 10: Line 157: The doses from Table 1 should be in the text of the “Dose-response experiments of florpyrauxifen-benzyl” session. It is not necessary a table to show the doses.

Response 10: Thanks to your kind reminder, we have combined the contents of the “Dose-response experiments of florpyrauxifen-benzyl” session and the “Florpyrauxifen-benzyl treatment” session but retained the use of tables to show the doses because the doses applied to R and S biotypes of barnyard grasses are different respectively, and it is more intuitive to show them in tables, as is the case in many other articles.

Point 11: Line 174: E. crus-galli should be italic

Response 11: Thank you for the correction, but we think there must be some misunderstanding here, because this error is the same as the Point 1, where the scientific name has been submitted in italics in the second version of the manuscript (agronomy-2185511_V2).

Point 12: Line 177: “UTR_1, UTR_2, UTR_3, UTS_1, UTS_2, UTS_3, FTR_1, FTR_2, FTR_3, 177 FTS_1, FTS_2, and FTS_3” = It is not necessary to show all replicate names. You just need to show the treatment name once.

Response 12: Thank you for the suggestion, but we think listing all the sample names would better correspond to the sample groupings in the original data, and also give the reader a better visualization of how we set up the groupings and replicates.

Point 13: Line 190: replace “Difffferential” by Differential

Response 13: Thank you for the correction. The misspelled words have been corrected.

Point 14: Line 190 - Session “Differential Gene Expression and Enrichment Analysis”: describe better the RNAseq analysis. Which software did you use to treat the data? Which software did you use to align the reads? To build the transcriptome? Which software did you use to blast with NR database? What Log2FC did you use as threshold?

Response 14: Thanks to your suggestion, we have added a session “Transcriptome Assembly and Functional Classification” to illustrate these issues you mentioned. (Line 203-215)

Point 15: Line 192: DEGseq2? Or DEseq2?

Response 15: We are very sorry, the correct one should be "DESeq2" and we have corrected it.

Point 16: Line 216 and 221: Replace qRT–PCR for RT-qPCR

Response 16: Thank you for your suggestion, all "qRT-PCR" has been replaced by "RT-qPCR" as suggested in point 21.

Point 17: Paragraph line 240 and paragraph line 243: you should build one paragraph. A paragraph cannot be only one sentence. The same problem for paragraphs from lines 256 and 259.

Response 17: Thanks to your kind reminder, we have adjusted the structure of the paragraphs and now all have at least two sentences in a paragraph.

Point 18: Line 290-295: Add space between the word and the (, for example: “ABA(abscisic acid)”

Response 18: Thank you for the correction. We have added spaces in the appropriate places.

Point 19: Line 301: The paragraph cannot be only one sentence.

Response 19: Thank you for the correction. We have noted the error you pointed out and have corrected it.

Point 20: Line 301: “Thirty-two” remove the capital letter.

Response 20: Thank you for the correction. We have noted the error you pointed out and have corrected it.

Point 21: Line 302 and 307: “Quantitative Real-Time PCR (qRT– PCR)” is wrong, the quantitative is the PCR, then the correct is RT-qPCR.

Response 21: Thank you for your correction, we are sorry because we saw other articles with abbreviations using both qRT- PCR or RT-qPCR, so we took it for granted that the two are common. It has been changed to RT-qPCR uniformly according to your comment. An example of other articles using the expression "qRT-PCR" can be seen at this link “DOI: 10.1186/s12943-022-01501-3”, and there are many more such expressions on Pubmed.

Point 22: Line 344-345: “The reason for the more intense auxin response in S may be related to IAA biosynthesis”: I think the main reason is because the R biotype degrades the herbicide faster, and avoid the auxin cascade effect.

Response 22: Thanks for your kind suggestion, we wanted to express that IAA biosynthesis is also one of the reasons. We have rewritten this paragraph of discussion and it looks better now. 

Point 23: Line 372 -376: this is results, not discussion.

Response 23: Thanks for your correction, but we think it should be allowed to cite results for discussion when necessary. We have rewritten this discussion paragraph and it looks better now.

Point 24: Line 399: You discuss a new gene GID1, that was not shown before.

Response 24: Thank you for your suggestion, but we did mention GID1 in the Supplementary Table S6, and to avoid unnecessary misunderstandings, we added "(Supplementary Table S6)" at the end of this sentence discussing GID1. 

Point 25: Line 432-437: The conclusion cannot finish like this.

Response 25: Thanks for the heads up, we rewrite the conclusion section and it looks much better now.

Point 26: Figure 1: The equation and parameters are too small.

Response 26: Thank you for your suggestion, we accepted it and enlarged the font size of the equations and parameters in the Figure 1.

Point 27: Figure 3: too small as well.

Response 27: Thanks for your suggestion, 4 images together did look a bit crowded and too small, so we split the up-regulated and down-regulated ones into two figures (Figure 3 and 4). 

Point 28: Figure 4: What is x axis?  and y axis?

Response 28: Thanks for your correction, we added the names of the horizontal and vertical axes in this version of the figure.

Point 29: Table 2: How and why did you choose these genes to analyze in qPCR? What about the other ~400 genes up- and down-regulated in both contrasts?

Response 29: Thanks for your kind suggestion, we think we should explain that in this paper the purpose of RT-qPCR is to validate the reliability of transcriptomic data and is not used for analysis. In some other transcriptome articles in the past usually about 10 to 20 genes were randomly selected for RT-qPCR validation, and in our study we also randomly selected many genes for RT-qPCR validation, but considering the content of the title and discussion section of this article, we thought that showing the results of RT-qPCR validation in relation to the title and discussion of the article for plant hormone signal transduction-related genes is necessary and more convincing.

Point 30: Figure 5: The genes shown in the heat map do not match the results shown in Table 2. For example, ACS is up-regulated in both contrasts, but in Figure 5 it is down-regulated for R contrast. ACO as well.

Response 30: We are very sorry for the misunderstanding about the color we used, according to our chart notes, not blue color necessarily means downward adjustment, the log2FC of the gene is also shown in blue color in the heatmap when it is in the range of 0 to 5. To avoid this misunderstanding, we have reset the color and variation range of the heatmap. 

Special thanks to you for your good comments.

We tried our best to improve the manuscript and made some changes in the manuscript. These changes will not influence the content and framework of the paper. And here we did not list all the changes but marked in colored words or colored background in revised paper.

We appreciate for your warm work earnestly, and hope that the correction will meet with approval.

Once again, thank you very much for your comments and suggestions.

Round 3

Reviewer 3 Report

Dear authors,

The manuscript improved when compared with the first version. However, I still have the opinion that it is not suitable to be published in Agronomy. I recommend you publish it in another journal. You performed RNAseq, but the paper content does not reach the required quality to publish in Agronomy. Please, review and describe in more detail the RNAseq analysis. For example: which software did they use to map the reads in the transcriptome? did they use salmon? what was the Log2FoldChange cut-off? I hope my comments can help to improve the manuscript's quality.

Author Response

Dear Professor:

Thank you for your review comments on our manuscript entitled "Comparative Transcriptome Analysis of the Differential Effects of Florpyrauxifen-benzyl Treatment on Phytohormone Transduction between Florpyrauxifen-benzyl-resistant and Susceptible Barnyard grasses (Echinochloa crus-galli (L.) P.Beauv)". (ID: agronomy-2185511). These comments are all valuable and very helpful in revising and improving our paper and of great importance to our research.

We have carefully studied the comments and made corrections that we hope will meet with approval.

The revised parts are indicated in the paper by colored words or colored background. The major corrections in the paper and the responses to the reviewers' comments are as follows:

Responds to the reviewe comments:

Point 1: which software did they use to map the reads in the transcriptome?

Response 1: Thank you for your kind correction. We added a description as “The transcriptome obtained by Trinity splicing was used as the reference sequence (Ref), and the clean reads of each sample were mapped to the Ref by RSEM software (v1.2.15), which uses the bowtie2 parameter mismatch 0 (the default bowtie2 parame-ter).”

Point 2:  what was the Log2FoldChange cut-off?

Response 2: Thank you for your kind reminder. We added a description as “Genes with an adjusted P-value <0.05 and |log2 (fold change)| > 1 found by DESeq2 were classified as differentially expressed. “

Special thanks to you for your good comments.

We tried our best to improve the manuscript and made some changes in the manuscript. These changes will not influence the content and framework of the paper. And here we did not list all the changes but marked in colored words or colored background in revised paper.

We appreciate for your warm work earnestly, and hope that the correction will meet with approval.

Once again, thank you very much for your comments and suggestions.
